# Antibiotic treatment duration for bloodstream infections in critically ill children—A survey of pediatric infectious diseases and critical care clinicians for clinical equipoise

Sandra Pong[1]*, Robert A. Fowler[2,3,4], Srinivas Murthy[5,6], Jeffrey M. Pernica[7], Elaine Gilfoyle[8], Patricia Fontela[9,10], Nicholas Mitsakakis[11,12], Asha C. Bowen[13,14], Winnie Seto[1,4,15], Michelle Science[16], James S. Hutchison[8], Philippe Jouvet[17,18], Asgar Rishu[19], Nick Daneman[20]

1 Department of Pharmacy, The Hospital for Sick Children, Toronto, Ontario, Canada, 2 Interdepartmental Division of Critical Care Medicine, University of Toronto, Toronto Ontario, Canada, 3 Tory Trauma Program, Sunnybrook Health Sciences Centre, Toronto, Ontario, Canada, 4 Institute of Health Policy, Management and Evaluation, University of Toronto, Toronto, Ontario, Canada, 5 Department of Pediatrics, Division of Critical Care, University of British Columbia, Vancouver, British Columbia, Canada, 6 Research Institute, BC Children's Hospital, Vancouver, British Columbia, Canada, 7 Division of Infectious Diseases, McMaster University, Hamilton, Ontario, Canada, 8 Department of Critical Care Medicine, The Hospital for Sick Children, Toronto, Ontario, Canada, 9 Department of Epidemiology, Biostatistics and Occupational Health, McGill University, Montreal, Québec, Canada, 10 Department of Pediatrics, McGill University, Montreal, Québec, Canada, 11 Children's Hospital of Eastern Ontario Research Institute, Ottawa, Ontario, Canada, 12 Dalla Lana School of Public Health, University of Toronto, Toronto, Ontario, Canada, 13 Wesfarmers Centre for Vaccines and Infectious Diseases, Telethon Kids Institute, University of Western Australia Perth Children's Hospital, Nedlands, Western Australia, Australia, 14 Department of Infectious Diseases, Perth Children's Hospital, Nedlands, Western Australia, Australia, 15 Leslie Dan Faculty of Pharmacy, University of Toronto, Toronto, Ontario, Canada, 16 Division of Infectious Diseases, Department of Paediatric Medicine, The Hospital for Children, Toronto, Ontario, Canada, 17 Pediatric Intensive Care Unit, Sainte-Justine Hospital University Center, Montreal, Québec, Canada, 18 Department of Pediatrics, Université de Montréal, Montreal, Québec, Canada, 19 Critical Care Research Unit, Sunnybrook Health Sciences Centre, Toronto, Ontario, Canada, 20 Division of Infectious Diseases, Sunnybrook Health Sciences Centre, Toronto, Ontario, Canada

* sandra.pong@sickkids.ca

**Data Availability Statement:** All relevant data are within the paper and its Supporting information files.

## Abstract

### Objective

To describe antibiotic treatment durations that pediatric infectious diseases (ID) and critical care clinicians usually recommend for bloodstream infections in critically ill children.

### Design

Anonymous, online practice survey using five common pediatric-based case scenarios of bloodstream infections.

### Setting

Pediatric intensive care units in Canada, Australia and New Zealand.

**Funding:** The authors received no specific funding for this work. Sandra Pong is supported by a SickKids Clinician-Scientist Training Program Scholarship from The Hospital for Sick Children.

**Competing interests:** The authors have declared that no competing interests exist.

## Participants

Pediatric intensivists, nurse practitioners, ID physicians and pharmacists.

## Main outcome measures

Recommended treatment durations for common infectious syndromes associated with bloodstream infections and willingness to enrol patients into a trial to study treatment duration.

## Results

Among 136 survey respondents, most recommended at least 10 days antibiotics for bloodstream infections associated with: pneumonia (65%), skin/soft tissue (74%), urinary tract (64%) and intra-abdominal infections (drained: 90%; undrained: 99%). For central vascular catheter-associated infections without catheter removal, over 90% clinicians recommended at least 10 days antibiotics, except for infections caused by coagulase negative staphylococci (79%). Recommendations for at least 10 days antibiotics were less common with catheter removal. In multivariable linear regression analyses, lack of source control was significantly associated with longer treatment durations (+5.2 days [95% CI: 4.4–6.1 days] for intra-abdominal infections and +4.1 days [95% CI: 3.8–4.4 days] for central vascular catheter-associated infections). Most clinicians (73–95%, depending on the source of bloodstream infection) would be willing to enrol patients into a trial of shorter versus longer antibiotic treatment duration.

## Conclusions

The majority of clinicians currently recommend at least 10 days of antibiotics for most scenarios of bloodstream infections in critically ill children. There is practice heterogeneity in self-reported treatment duration recommendations among clinicians. Treatment durations were similar across different infectious syndromes. Under appropriate clinical conditions, most clinicians would be willing to enrol patients into a trial of shorter versus longer treatment for common syndromes associated with bloodstream infections.

## Introduction

Bloodstream infections (BSIs) cause significant morbidity and mortality in critically ill pediatric patients. In a three-year surveillance of BSIs in a pediatric intensive care unit, there were 39 cases of BSIs per 1000 admissions, and these patients experienced a crude mortality three-fold higher than uninfected patients [1]. Optimal treatment of infections involves both timely initiation of adequate treatment [2–4] as well as continuation of treatment for an appropriate duration. However, the optimal treatment duration with antimicrobials for BSIs has not been determined [5–8]. Shortened treatment durations might offer potential benefits in patient-centred outcomes including earlier hospital discharge and reduced antimicrobial adverse events, *C. difficile* infections, antimicrobial resistance and healthcare costs. Therefore, it is important to know whether shorter treatment durations would lead to similar clinical cure and survival. A trial of shorter versus longer treatment duration for bacteremia is underway for critically ill adults [9]. Given the paucity of evidence guiding current practices on treatment duration, a

trial among pediatric patients may be warranted, in order to explore generalizability and examine pediatric-relevant outcomes.

Previous surveys of critical care and infectious diseases (ID) physicians in Canada, Australia and New Zealand have reported wide practice variation in recommended antibiotic treatment durations for five common bacteremic syndromes among adults: pneumonia, skin/soft tissue, urinary tract, intra-abdominal and central vascular catheter-associated infections. They found that the most common treatment durations recommended were 7, 10 and 14 days and most respondents would not modify their recommendations about treatment duration based on host characteristics or measures of clinical response [10,11]. Pediatric clinicians generally view infants and children as a distinct group from adults and self-reported treatment recommendations for adults may not reflect practices for infants and children. Pediatric patients can experience different disease and developmental issues, and potentially respond differently to medical therapies [2,12–14].

To examine pediatric-specific BSI treatment duration recommendations and explore potential factors that might influence clinical decisions on treatment, we conducted a survey among pediatric ID physicians, critical care clinicians and pharmacists using pediatric-based case scenarios. The objectives of this survey study were to describe the self-reported durations of antibiotic therapy that pediatric ID and critical care clinicians usually recommend for treatment of BSIs in critically ill children, and to determine if they would be willing to enrol such patients into a trial comparing shorter versus longer antibiotic treatment. We hypothesized that there would be practice heterogeneity among clinicians but that the majority of respondents are currently recommending longer treatment courses of at least 10 days for each of the common bacteremic syndromes.

## Methods

### Study setting and population

We conducted an anonymous, online practice survey via SurveyMonkey among pediatric ID and critical care clinicians in Canada, Australia and New Zealand between December 2020 and February 2021. Critical care clinicians and pharmacists in Canadian pediatric intensive care units were contacted by email with invitations to participate in the survey. The ID clinicians surveyed in Canada belonged to the Pediatric Investigators Collaborative Network on Infections in Canada (PICNIC), and those in Australia and New Zealand belonged to the Australia and New Zealand Paediatric Infectious Diseases Group (ANZPID) of the Australasian Society of Infectious Diseases (ASID). Respondents were provided with a link to the survey webpage for further study information and were requested to provide informed consent on the first page before they proceeded to the next page to start answering the survey questions. The survey was conducted anonymously online, and participants could withdraw participation at any time by closing the survey prior to final submission. Continuation beyond the first page of the survey, followed by completion and submission of the survey constituted informed consent for participation. Two reminder email invitations were sent after the initial invitation to participate in the survey. Research Ethics Board approval for the survey and consent process was granted at Sunnybrook Health Sciences Centre, University of Toronto (Toronto, Canada).

### Survey design

The survey was based on a survey tool previously used in a Canadian study conducted with clinicians caring for critically ill adults [10] but modified to include cases with a pediatric focus, suitable for pediatric clinicians. The survey consisted of 5 pediatric scenarios describing

common infectious syndromes associated with BSIs in the pediatric intensive care unit: pneumonia, skin/soft tissue, urinary tract, intra-abdominal and central vascular catheter-associated infections. For each scenario, clinicians were asked what total duration of antibiotic therapy they would usually recommend for the patient. In the case of bacteremia involving a central vascular catheter, clinicians were asked to specify their recommendations if the causative pathogens were: *Enterococcus faecalis*, *Staphylococcus aureus*, *Klebsiella pneumoniae*, coagulase negative staphylococci, *Escherichia coli*, *Enterobacter cloacae* and *Pseudomonas aeruginosa*. For every case scenario, survey respondents were also asked whether they would be willing to enrol a patient like the one described in the case into a study comparing 7 versus 14 days of antibiotic therapy. After the 5 pediatric scenarios, respondents were provided a list of patient outcome measures and asked to indicate which ones they considered important trial outcomes that would influence their antibiotic treatment practices for critically ill children with BSIs. They were also asked whether they would be willing to shorten their treatment duration recommendations based on evidence extrapolated from an adult trial demonstrating that the mortality rate in adults receiving shorter duration therapy was no worse than adults receiving longer duration therapy.

The survey was pilot-tested by 2 ID physicians, 3 critical care physicians, 1 critical care nurse practitioner and 1 critical care pharmacist. They assessed the flow, acceptability and ease of administration of the survey tool, and the clarity and interpretation of the questions [15]. After pilot-testing, two additional questions regarding the existence and participation in antimicrobial stewardship programs (ASP) were added. The final survey included 5 scenarios and 33 items (S1 File).

## Sample size

The target sample size for the survey was 97 respondents to allow a 95% two-sided confidence interval to extend ±10% around an expected proportion of 50% of respondents that would recommend a longer course (e.g. ≥10 days) versus a shorter course of antimicrobial therapy for any specific infectious syndrome ($\alpha = 0.05$) [10,16].

## Statistical analyses

We described recommended antibiotic treatment durations using frequencies and percentages, medians and interquartile ranges and graphical displays of the responses. We defined longer treatment duration as ≥10 days of antimicrobial therapy [17]. Univariate analyses (Wilcoxon Rank Sum Test for 2 groups and Kruskal-Wallis Test for >2 groups) were performed to determine if clinician characteristics (practice specialty, country of practice, years of experience and ASP activity) were associated with recommended treatment durations. The willingness of clinicians to enrol patients into a trial of 7 versus 14 days of antimicrobials, and to shorten their treatment duration recommendations based on evidence extrapolated from an adult trial were described using frequencies and percentages. Clinical trial outcomes that would influence clinicians' practices on treatment duration were counted and graphically illustrated. We conducted multivariable linear regression analyses with treatment duration for each bacteremia syndrome as the dependent variable and clinician specialty, country of practice, years since graduation and ASP at practice site as independent variables. For the intra-abdominal infection scenario, source control was included as an additional predictor variable. For the central vascular catheter-associated infection scenario, catheter removal and pathogen type were additional predictor variables. Statistical analyses were conducted using SAS statistical software version 9.4M6 (SAS Institute, Cary, NC) and R version 4.0.2.

## Results

### Characteristics of survey respondents

There were 136 survey respondents, including 77 (57%) critical care clinicians (intensivists, fellows and nurse practitioners), 32 (24%) ID physicians, 20 (15%) critical care pharmacists and 7 (5%) clinicians with combined training in critical care, ID, pediatrics, pharmacy, surgery, cardiology or clinical microbiology. The overall response rate was 26%—with higher response rates among pharmacists (51%) and critical care clinicians (50%) and lower response rates among ID physicians (16%). All respondents practiced in academic institutions and 97% treated only pediatric patients. The respondents had a broad range of clinical experience: ≤5 years (7%), to 6–10 years (24%), 11–15 years (23%), 16–20 years (22%) and ≥21 years (24%) of clinical practice. Half (50%) of the respondents reported managing more than 20 cases of BSIs per year. Most respondents were clinicians who were currently practicing in Canada (85%), the remainder were practicing in Australia and New Zealand (15%). There was an ASP program established in 75% of respondents' institutions. Among those with ASP programs, 42% of respondents reported being an active member of the ASP team.

### Treatment duration recommendations

The majority of clinicians recommended at least 10 days of antibiotics for bacteremia associated with each of the infectious syndromes: pneumonia (84/129, 65%), skin/soft tissue (80/108, 74%), urinary tract (68/107, 64%), intra-abdominal (drained) (97/108, 90%) and intra-abdominal (undrained) (107/108, 99%). The most common treatment durations recommended for most infections were either 7, 10 or 14 days (full distributions displayed in Figs 1 and 2). Treatment durations of 21 days or longer were sometimes recommended for undrained intra-abdominal infections and central vascular catheter-associated infections without catheter removal.

For central vascular catheter-associated BSIs where the catheter was not removed, over 90% of clinicians recommended at least 10 days of antibiotics for all pathogens, except for coagulase negative staphylococci (86/109, 79%). When the infected catheter was removed, 64 to 78% (depending on the pathogen) of clinicians still recommended at least 10 days of antibiotics irrespective of the causative pathogen, except for *E. faecalis* (53/110, 48%) and coagulase negative staphylococci (35/110, 32%).

Table 1 summarizes the overall recommended treatment durations by infectious syndrome. There were significant differences in median recommended treatment durations depending on whether intravenous catheters were removed or not (10 [IQR 7–14] days vs. 14 [IQR 14–14] days) and whether intra-abdominal infections were drained or only partially/not drained (14 [IQR 10–14] days vs. 21 [IQR 14–21] days).

Overall, there were no significant differences in treatment durations recommended by critical care clinicians, ID physicians and pharmacists for bacteremia associated with pneumonia, skin/soft tissue, urinary tract and intra-abdominal infections (S1 Table). For central vascular catheter-associated bacteremia with source control (infected catheter removed), ID physicians recommended significantly shorter treatment durations than critical care clinicians and pharmacists for *E. faecalis* and coagulase negative staphylococci infections. When there was no removal of infected catheters, ID physicians recommended significantly shorter durations than critical care clinicians and pharmacists for all pathogens, except for *S. aureus*, *K. pneumoniae* and *P. aeruginosa* infections (Table 2).

Recommendations for treatment durations for the various bacteremia syndromes in the survey scenarios were generally similar between clinicians based on years of clinical practice,

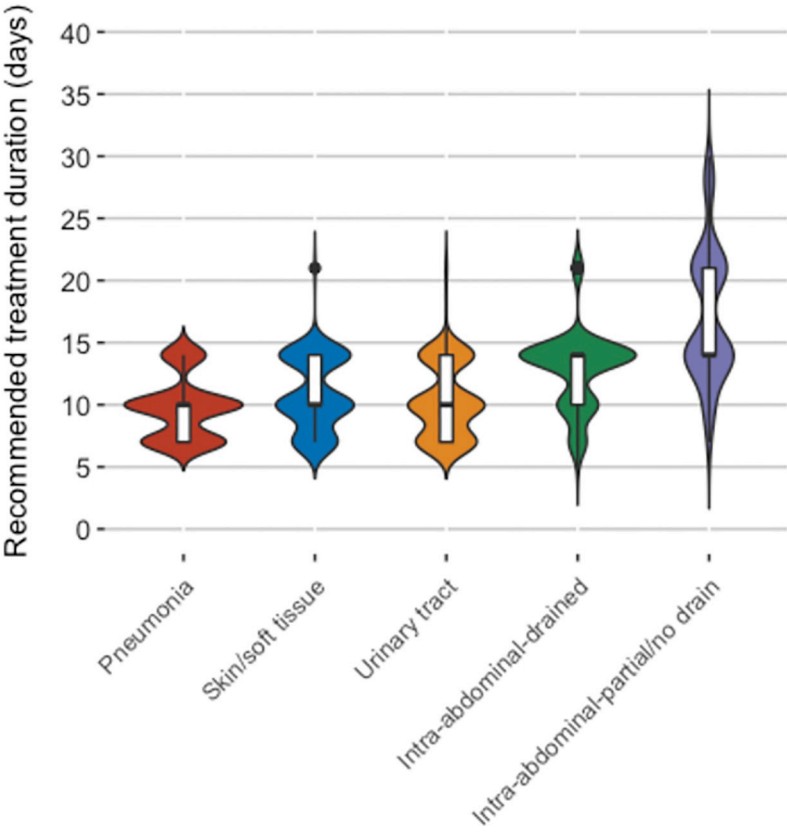

**Fig 1. Violin plot distribution of recommended treatment durations for bacteremia from different sources.** The shaded area represents the density of responses for treatment duration. Inside each violin, the thick line marks the median, the boxes indicate the interquartile range (IQR) and the lines mark 1.5xIQR for the treatment duration recommendation.

the number of BSIs managed per year and whether an ASP program was established at practice sites or not (S2–S4 Tables).

## Multivariable predictors of recommended treatment durations

No association was found between clinician specialty and recommended treatment duration for the bacteremic syndromes presented in the survey scenarios, except for skin/soft tissue-related infections where critical care clinicians recommended 2.3 days (95% CI: -3.8 to -0.8 days) shorter treatment compared to ID clinicians (Table 3). For bacteremia associated with pneumonia, clinicians in Canada recommended 1.7 days (95% CI: 0.3 to 3.2 days) longer treatment than clinicians in Australia and New Zealand. Recommended treatment duration of skin/soft tissue-related bacteremia was also 3.4 days (95% CI: 1.6 to 5.2 days) longer among Canadian clinicians (Table 3).

Treatment durations were 5.2 days (95% CI: 4.4 to 6.1 days) longer for undrained intra-abdominal infections and 4.1 days (95% CI: 3.8 to 4.4 days) longer for central vascular catheter-associated infections without catheter removal. The type of pathogen in central vascular catheter-associated BSIs was also associated with treatment duration recommendations (Table 3).

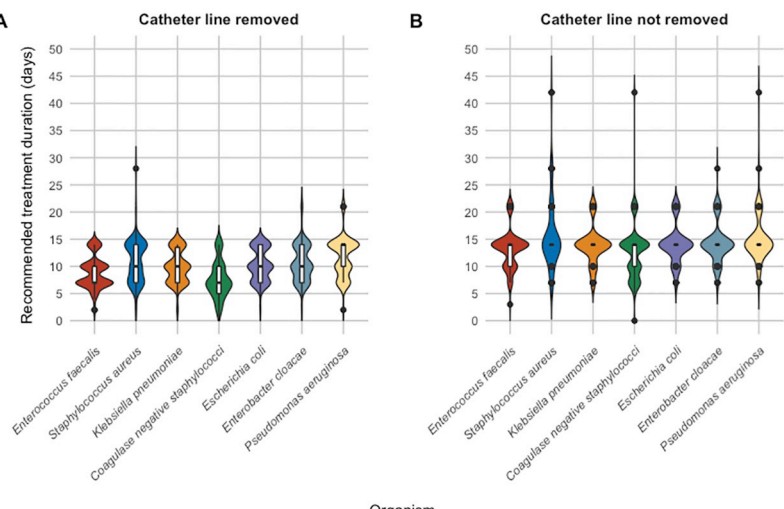

**Fig 2. Violin plot distribution of recommended treatment durations for central vascular catheter-related bacteremia caused by different pathogens.**

## Willingness to enrol patients into a trial of shorter versus longer treatment duration

Most clinicians would be willing to enrol patients with bacteremia into a trial of 7 versus 14 days of antibiotics, when the bacteremia is associated with pneumonia (109/129, 84%), skin/ soft tissue (85/108, 79%), urinary tract (97/107, 91%), intra-abdominal infections (drained) (79/108, 73%) or central vascular catheter-associated infections (105/110, 95%). For this later scenario, 61% (64/105) would be willing regardless of catheter removal, 35% (37/105) would be willing only if the catheter was removed and 4% (4/105) would be willing only if the catheter was not removed.

The majority of clinicians who were willing to enrol patients into a trial of 7 versus 14 days of antibiotics made recommendations for long treatment durations of at least 10 days for bacteremia in the case scenarios: pneumonia (79/109, 72%), skin/soft tissue (64/85, 75%), urinary tract (64/97, 66%) and drained intra-abdominal infections (70/79, 89%). For central vascular catheter-associated infections with catheter removal, depending on the pathogen, 63 to 77% of clinicians who would enrol patients in a trial recommended at least 10 days of antibiotics for all the pathogens listed (S5 Table), except for *E. faecalis* (48/105, 46%) and coagulase negative staphylococci (30/105, 29%). Without catheter removal, over 90% of clinicians who would enrol patients in a trial had recommended at least 10 days of antibiotics for various pathogens, except for coagulase negative staphylococci (81/104, 78%) (S5 Table).

## Clinical trial outcomes in pediatrics and extrapolation of adult data

Fig 3 shows that more than 50% of respondents indicated that measures of organ dysfunction, time to clinical stability, length of stay in the intensive care unit, duration of mechanical ventilation and length of hospital stay were important outcome measures in a pediatric trial. In contrast, neurodevelopmental outcomes, functional status outcomes, measures of frailty and quality of life were selected by fewer than 25% of respondents. Most clinicians indicated that they would be willing to extrapolate results from an adult trial comparing 7 versus 14 days of antibiotic treatment for bacteremia to their own pediatric practices (84/107, 79%).

**Table 1. Recommended treatment duration (days) by infectious syndrome.**

| Infectious syndrome | n | Median | IQR | Range |
|---|---|---|---|---|
| **Pneumonia** | | | | |
| | 129 | 10 | 7–10 | 7–14 |
| **Skin/soft tissue** | | | | |
| | 108 | 10 | 7.5–14 | 7–21 |
| **Urinary tract** | | | | |
| | 107 | 10 | 7–14 | 7–14 |
| **Intra-abdominal**[a] | | | | |
| **Drained** | 108 | 14 | 10–14 | 5–21 |
| **Partial/not drained** | 108 | 21 | 14–21 | 7–30 |
| **Central vascular catheter-associated**[b] | | | | |
| **Catheter removed (overall)** | 110 | 10 | 7–14 | 0–28 |
| *Enterococcus faecalis* | | 7 | 7–10 | 2–14 |
| *Staphylococcus aureus* | | 10 | 7–14 | 2–28 |
| *Klebsiella pneumoniae* | | 10 | 7–14 | 2–14 |
| *Coagulase negative staphylococci* | | 7 | 5–10 | 0–14 |
| *Escherichia coli* | | 10 | 7–14 | 2–14 |
| *Enterobacter cloacae* | | 10 | 7–14 | 2–21 |
| *Pseudomonas aeruginosa* | | 14 | 10–14 | 2–21 |
| **Catheter not removed (overall)** | 109 | 14 | 14–14 | 0–42 |
| *Enterococcus faecalis* | | 14 | 10–14 | 3–21 |
| *Staphylococcus aureus* | | 14 | 14–14 | 7–42 |
| *Klebsiella pneumoniae* | | 14 | 14–14 | 7–21 |
| *Coagulase negative staphylococci* | | 14 | 10–14 | 0–42 |
| *Escherichia coli* | | 14 | 14–14 | 7–21 |
| *Enterobacter cloacae* | | 14 | 14–14 | 7–28 |
| *Pseudomonas aeruginosa* | | 14 | 14–14 | 7–42 |

[a]Difference between intra-abdominal infection drained vs. partial/no drain: p<0.05 (Wilcoxon Rank Sum Test).

[b]Difference between catheter removed (overall) vs. catheter not removed (overall): p<0.05 (Wilcoxon Rank Sum Test).

## Discussion

In our survey of pediatric ID clinicians, critical care clinicians and pharmacists in Canada, Australia and New Zealand, we found that at least 60% of clinicians would recommend 10 days or longer of antibiotics when treating bacteremia associated with pneumonia, skin/soft tissue, urinary tract, intra-abdominal and central vascular catheter-associated infections. The recommended treatment durations and variability were similar across different infectious syndromes and between clinician specialties, supporting implicit collective equipoise for a study of shorter versus longer duration treatment for BSIs in critically ill children. Pending availability of evidence-based criteria to guide treatment duration of BSIs in children, most clinicians indicated that they would be willing to apply adult trial data to their pediatric practices. Under appropriate clinical conditions, most clinicians would be willing to enrol patients into a trial of 7 versus 14 days of antimicrobial treatment for different bacteremic syndromes. Interestingly, the majority of clinicians who would enrol patients into such a trial also recommended longer antibiotic treatment durations for at least 10 days in the survey scenarios. This suggests that they recognize that shorter duration therapy could be appropriate for some patients, despite their current self-reported practices. We hypothesize that this dissonance exists because

**Table 2. Median (IQR) treatment duration (days) of central vascular catheter-associated bacteremia by clinician specialty.**

|  | Critical care (n = 68) | Infectious diseases (n = 29) | Pharmacy (n = 13) | p-value[a, b] |
|---|---|---|---|---|
| **Catheter removed (n = 110)** |  |  |  |  |
| *Enterococcus faecalis* | 10 (7–10) | 7 (7–10) | 7 (7–10) | 0.004 |
| *Staphylococcus aureus* | 10 (7–14) | 14 (7–14) | 10 (7–14) | 0.31 |
| *Klebsiella pneumoniae* | 10 (7–14) | 10 (7–10) | 7 (7–10) | 0.31 |
| **Coagulase negative staphylococci** | 7 (7–10) | 5 (3–7) | 7 (7–7) | <0.001 |
| *Escherichia coli* | 10 (8.5–14) | 10 (7–10) | 10 (7–10) | 0.2 |
| **Enterobacter cloacae** | 10 (10–14) | 10 (7–10) | 10 (7–10) | 0.07 |
| *Pseudomonas aeruginosa* | 14 (10–14) | 10 (7–14) | 10 (7–14) | 0.21 |
| **Catheter not removed (n = 109)** |  |  |  |  |
| *Enterococcus faecalis* | 14 (14–14)[c] | 10 (10–14) | 14 (10–14) | <0.001 |
| *Staphylococcus aureus* | 14 (14–21)[c] | 14 (14–14) | 14 (14–21) | 0.55 |
| *Klebsiella pneumoniae* | 14 (14–14)[c] | 14 (10–14) | 14 (14–14) | 0.03 |
| **Coagulase negative staphylococci** | 14 (10–14)[c] | 10 (7–10) | 14 (10–14) | <0.001 |
| *Escherichia coli* | 14 (14–14)[c] | 14 (10–14) | 14 (14–14) | 0.009 |
| **Enterobacter cloacae** | 14 (14–14)[c] | 14 (10–14) | 14 (14–14) | 0.005 |
| *Pseudomonas aeruginosa* | 14 (14–21)[c] | 14 (14–14) | 14 (14–14) | 0.04 |

[a]Kruskal-Wallis Test.

[b]Bonferroni adjusted p-value threshold = 0.02.

[c]Missing = 1.

previous guidelines relating to antibiotic management of bacteremic syndromes endorsed these longer durations [18,19]; however, recent evidence has demonstrated that shorter-duration therapy is preferable for a number of different infectious syndromes [8,20–25]. In addition, earlier switch from intravenous to suitable oral antibiotics with patient defervescence and clinical improvement is another potential strategy to optimize treatment duration and reduce antimicrobial overuse [8]. Clinicians may realize that their current practices are overly prolonged but need evidence to support modifications to management guidelines. Our results demonstrate explicit support for equipoise to enrol pediatric patients into a trial that could potentially help to decrease unnecessary antimicrobial use and prevent harms associated with antibiotic overuse. While outcomes like organ dysfunction, clinical stability, length of stay and duration of mechanical ventilation are objective trial outcome measures that could influence clinician practices, additional outcomes like neurodevelopmental milestones, functional status, quality of life and frailty measures that are important to patients and their families should also be considered.

The variability in antibiotic duration recommendations found in our survey is similar to patterns found among adult ID and critical care clinicians [10,11]. However, in contrast to the adult results, our survey of pediatric clinicians did not detect a difference between ID and critical care clinicians in their recommendations for treatment durations for bacteremia associated with pneumonia, skin/soft tissue, urinary tract and intra-abdominal infections. Recommendations by pediatric pharmacists were also similar to ID and critical care clinicians.

In our regression models of treatment duration recommendations for bacteremia associated with pneumonia, skin/soft tissue and urinary tract infections, we found that clinicians in Australia/New Zealand recommended significantly shorter treatment durations for BSIs associated with pneumonia and skin/soft tissue infections compared to Canadian clinicians. However, the overall number of respondents in our subgroups were small and this difference

**Table 3. Multivariable regression models for predictors of recommended treatment duration for different infectious syndromes.**

| Predictor | Adjusted β-coefficient[a] | 95% confidence interval | p-value[b] |
|---|---|---|---|
| **PNEUMONIA (Omnibus F-test: 1.28 (8, 120), p = 0.26, R-square = 0.08)** | | | |
| **Specialty** | | | |
| **Pharmacist** | -0.8 | -2.3 to 0.6 | 0.26 |
| **Critical care** | -1 | -2.2 to 0.2 | 0.09 |
| **Infectious diseases** | Reference | - - | - - |
| **Country** | | | |
| **Canada** | 1.7 | 0.3 to 3.2 | 0.02 |
| **Australia/New Zealand** | Reference | - - | - - |
| **Years since graduation** | | | |
| **0–5** | -0.6 | -2.4 to 1.1 | 0.46 |
| **6–10** | 0.6 | -0.6 to 1.9 | 0.32 |
| **11–15** | -0.1 | -1.4 to 1.1 | 0.85 |
| **16–20** | 0.3 | -0.9 to 1.6 | 0.58 |
| **≥21** | Reference | - - | - - |
| **Antimicrobial stewardship program** | | | |
| **Yes** | -0.3 | -1.3 to 0.7 | 0.55 |
| **No** | Reference | - - | - - |
| **SKIN/SOFT TISSUE (Omnibus F-test: 2.42 (8, 99), p <0.05, R-square = 0.16)** | | | |
| **Specialty** | | | |
| **Pharmacist** | 0.1 | -1.8 to 2 | 0.91 |
| **Critical care** | -2.3 | -3.8 to -0.8 | 0.004 |
| **Infectious diseases** | Reference | - - | - - |
| **Country** | | | |
| **Canada** | 3.4 | 1.6 to 5.2 | <0.001 |
| **Australia/New Zealand** | Reference | - - | - - |
| **Years since graduation** | | | |
| **0–5** | 0.4 | -2.1 to 3 | 0.73 |
| **6–10** | -0.4 | -1.9 to 1.2 | 0.65 |
| **11–15** | -0.5 | -2 to 1.1 | 0.54 |
| **16–20** | 0.1 | -1.5 to 1.6 | 0.92 |
| **≥21** | Reference | - - | - - |
| **Antimicrobial stewardship program** | | | |
| **Yes** | 0.2 | -1.1 to 1.5 | 0.77 |
| **No** | Reference | - - | - - |
| **URINARY TRACT (Omnibus F-test: 0.65 (8, 98), p = 0.74, R-square = 0.05)** | | | |
| **Specialty** | | | |
| **Pharmacist** | -1 | -2.9 to 0.9 | 0.31 |
| **Critical care** | -1.1 | -2.6 to 0.4 | 0.15 |
| **Infectious diseases** | Reference | - - | - - |
| **Country** | | | |
| **Canada** | 0.8 | -1 to 2.6 | 0.39 |
| **Australia/New Zealand** | Reference | - - | - - |
| **Years since graduation** | | | |
| **0–5** | -0.7 | -3.3 to 1.8 | 0.56 |
| **6–10** | 0.7 | -0.9 to 2.2 | 0.4 |
| **11–15** | -0.4 | -1.9 to 1.2 | 0.64 |
| **16–20** | 0.1 | -1.4 to 1.7 | 0.88 |

*(Continued)*

**Table 3.** (Continued)

| Predictor | Adjusted β-coefficient[a] | 95% confidence interval | p-value[b] |
|---|---|---|---|
| ≥21 | Reference | – – | – – |
| **Antimicrobial stewardship program** | | | |
| Yes | -0.6 | -1.9 to 0.8 | 0.39 |
| No | Reference | – – | – – |
| **INTRA-ABDOMINAL (AIC = 1187.6)** | | | |
| **Specialty** | | | |
| **Pharmacist** | -2.2 | -4.6 to 0.3 | 0.08 |
| **Critical care** | -0.8 | -2.6 to 1.1 | 0.43 |
| **Infectious diseases** | Reference | – – | – – |
| **Country** | | | |
| **Canada** | 1.3 | -1 to 3.6 | 0.26 |
| **Australia/New Zealand** | Reference | – – | – – |
| **Years since graduation** | | | |
| **0–5** | -0.04 | -3.2 to 3.1 | 0.98 |
| **6–10** | -0.8 | -2.7 to 1.2 | 0.45 |
| **11–15** | -1.4 | -3.3 to 0.5 | 0.15 |
| **16–20** | -0.3 | -2.3 to 1.6 | 0.73 |
| **≥21** | Reference | – – | – – |
| **Antimicrobial stewardship program** | | | |
| **Yes** | -0.5 | -2.2 to 1.2 | 0.57 |
| **No** | Reference | – – | – – |
| **Source control** | | | |
| **Partial/no drainage** | 5.2 | 4.4 to 6.1 | <0.001 |
| **Drained** | Reference | – – | – – |
| **CENTRAL VASCULAR CATHETER-ASSOCIATED (AIC = 7991.9)** | | | |
| **Specialty** | | | |
| **Pharmacist** | 0.6 | -1 to 2.3 | 0.45 |
| **Critical care** | 1.1 | -0.2 to 2.3 | 0.09 |
| **Infectious diseases** | Reference | – – | – – |
| **Country** | | | |
| **Canada** | 1.4 | -0.2 to 2.9 | 0.08 |
| **Australia/New Zealand** | Reference | – – | – – |
| **Years since graduation** | | | |
| **0–5** | 2.6 | 0.4 to 4.7 | 0.02 |
| **6–10** | 1.2 | -0.1 to 2.5 | 0.08 |
| **11–15** | 1.1 | -0.2 to 2.4 | 0.1 |
| **16–20** | 1.4 | 0.1 to 2.7 | 0.03 |
| **≥21** | Reference | – – | – – |
| **Antimicrobial stewardship program** | | | |
| **Yes** | -0.5 | -1.6 to 0.6 | 0.39 |
| **No** | Reference | – – | – – |
| **Source control** | | | |
| **Catheter not removed** | 4.1 | 3.8 to 4.4 | <0.001 |
| **Catheter removed** | Reference | – – | – – |
| **Pathogen** | | | |
| *E. faecalis* | -2.4 | -3 to -1.9 | <0.001 |
| *S. aureus* | -0.2 | -0.7 to 0.4 | 0.59 |

*(Continued)*

**Table 3.** (Continued)

| Predictor | Adjusted β-coefficient[a] | 95% confidence interval | p-value[b] |
|---|---|---|---|
| *K. pneumoniae* | -1.6 | -2.2 to -1.1 | <0.001 |
| **Coagulase negative staphylococci** | -3.4 | -4 to -2.8 | <0.001 |
| *E. coli* | -1.3 | -1.9 to -0.7 | <0.001 |
| *E. cloacae* | -1 | -1.6 to -0.4 | 0.001 |
| *P. aeruginosa* | Reference | - - | - - |

[a]Adjusted β-coefficient for each predictor variable represents the change in number of days of antimicrobial therapy for the variable relative to the reference predictor variable.

[b]Bonferroni adjusted p-value thresholds = 0.02 for 'Specialty', 0.005 for 'Years since graduation', 0.002 for 'Pathogens'.

among countries may be confounded by survey respondents in Australia and New Zealand being primarily ID clinicians. Lack of source control could increase the potential for continued seeding of infection—this was a significant predictor of longer treatment duration for intra-abdominal and central vascular catheter-associated bacteremia, as was the type of causative pathogen in catheter-related infections. The presence of an institutional antimicrobial stewardship program was not associated with a significant difference in treatment duration recommendations made by survey respondents.

Limitations of this survey study include the overall low response rate and inherent biases of self-reporting in survey studies (as compared to actual practice). Survey responses on self-reported practices may not be an accurate reflection of actual clinical practice. We also acknowledge that only ID clinicians were surveyed in Australia and New Zealand via ANZPID and the ID physicians surveyed in Canada all belonged to PICNIC so respondents may not be representative of all ID clinicians in pediatric practices. Furthermore, three quarters of survey respondents had established ASP programs at their institutions and almost half of them were

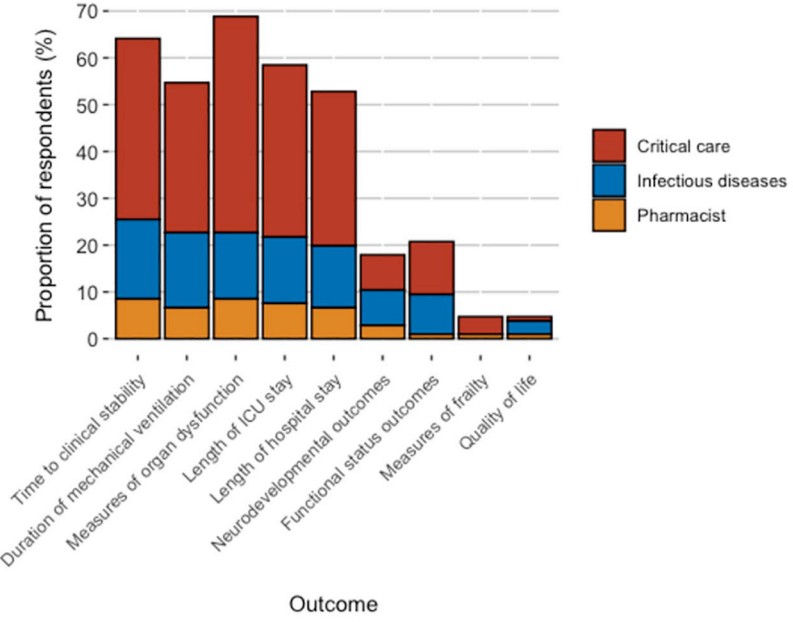

**Fig 3. Outcomes in pediatric trials that could influence antibiotic treatment practices.**

active members of ASP teams. Therefore, participants in our survey might have been more inclined to support ASP-related initiatives like reducing unnecessary antibiotic use and the actual median treatment durations that would be recommended by ID and critical care clinicians could be longer than what we detected.

## Conclusions

The results of this survey study confirm that there is variability in recommended antibiotic treatment durations for critically ill pediatric patients with BSIs, but most clinicians recommend treating for 10 days or longer. While pediatric ID and critical care clinicians would consider extrapolating results from an adult trial, we also demonstrated both implicit and explicit clinical equipoise among clinicians to enrol their own pediatric patients into a trial of 7 versus 14 days of antibiotic treatment for bacteremia. Ultimately, a generalizable trial of shorter versus longer antibiotic treatment is likely needed to inform treatment practices, so that the benefits of antibiotic therapy can be optimized while the harms are minimized for critically ill children.

## Supporting information

**S1 File. Survey document.**
(PDF)

**S1 Table. Median (IQR) treatment duration (days) by clinician specialty.**
(DOCX)

**S2 Table. Median (IQR) treatment duration (days) by years since graduation.**
(DOCX)

**S3 Table. Median (IQR) treatment duration (days) by number of bloodstream infections managed per year.**
(DOCX)

**S4 Table. Median (IQR) treatment duration (days) by presence of institutional antimicrobial stewardship program.**
(DOCX)

**S5 Table. Respondents who were willing to enrol patients into a trial of 7 versus 14 days of antimicrobial therapy and recommended at least 10 days of antimicrobial therapy for central vascular catheter-associated infections in case scenarios.**
(DOCX)

**S6 Table. Comparison of respondents recommending longer (≥10 days) treatment durations between infectious syndromes.**
(DOCX)

**S1 Dataset.**
(XLSX)

## Acknowledgments

We acknowledge the Pediatric Investigators Collaborative Network on Infections in Canada (PICNIC) and the Australia and New Zealand Paediatric Infectious Diseases Group (ANZPID) of the Australasian Society of Infectious Diseases (ASID) for graciously distributing the survey.

## Author Contributions

**Conceptualization:** Sandra Pong, Robert A. Fowler, Nick Daneman.

**Data curation:** Sandra Pong.

**Formal analysis:** Sandra Pong.

**Investigation:** Sandra Pong.

**Methodology:** Sandra Pong, Robert A. Fowler, Srinivas Murthy, Jeffrey M. Pernica, Elaine Gilfoyle, Patricia Fontela, Nicholas Mitsakakis, Asha C. Bowen, Winnie Seto, Michelle Science, James S. Hutchison, Philippe Jouvet, Asgar Rishu, Nick Daneman.

**Project administration:** Sandra Pong.

**Resources:** Sandra Pong.

**Software:** Sandra Pong.

**Supervision:** Robert A. Fowler, Nick Daneman.

**Validation:** Sandra Pong.

**Visualization:** Sandra Pong.

**Writing – original draft:** Sandra Pong.

**Writing – review & editing:** Sandra Pong, Robert A. Fowler, Srinivas Murthy, Jeffrey M. Pernica, Elaine Gilfoyle, Patricia Fontela, Nicholas Mitsakakis, Asha C. Bowen, Winnie Seto, Michelle Science, James S. Hutchison, Philippe Jouvet, Asgar Rishu, Nick Daneman.

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
