## [Decision Letter · Decision Letter 0]

19 May 2022

PONE-D-22-04577Antibiotic treatment duration for bacteremia in critically ill children--a survey of pediatric infectious diseases and critical care clinicians for clinical equipoisePLOS ONE

Dear Dr. Pong,

Thank you for submitting your manuscript to PLOS ONE. After careful consideration, we feel that it has merit but does not fully meet PLOS ONE’s publication criteria as it currently stands. Therefore, we invite you to submit a revised version of the manuscript that addresses the points raised during the review process.

While paper is one of much interest, the analyses of the data obtained requires more thought on the results and inferences. The reviewers have provided clear comments to be addressed which the authors are required to respond to.

We look forward to receiving your revised manuscript.

Kind regards,

Jamunarani Vadivelu

Academic Editor

PLOS ONE

Journal Requirements:

Additional Editor Comments:

While paper is one of much interest, the analyses of the data obtained requires more thought on the results and inferences. The reviewers have provided clear comments to be addressed which the authors are required to respond to.

Reviewers' comments:

Reviewer's Responses to Questions

**Comments to the Author**

1. Is the manuscript technically sound, and do the data support the conclusions?

Reviewer #1: Yes

Reviewer #2: Yes

Reviewer #3: Yes

2. Has the statistical analysis been performed appropriately and rigorously? 

Reviewer #1: No

Reviewer #2: Yes

Reviewer #3: Yes

3. Have the authors made all data underlying the findings in their manuscript fully available?

Reviewer #1: No

Reviewer #2: Yes

Reviewer #3: Yes

4. Is the manuscript presented in an intelligible fashion and written in standard English?

Reviewer #1: Yes

Reviewer #2: Yes

Reviewer #3: Yes

5. Review Comments to the Author

Reviewer #1: Review of article entitled “Antibiotic treatment duration for bacteremia in critically ill children--a survey of pediatric infectious diseases and critical care clinicians for clinical equipoise” by S. Pong et al.

Major comments

Interesting study. The project is well designed and the results are presented clearly. The use of violin plots is judicious. I have the following comments to help improve the manuscript.

1. Abstract: could we indicate how many scenarios were presented? Perhaps mentioning that the survey focused on 5 common pediatric scenarios.

2. Respondents: could an individual answer the survey twice?

3. Invitation to participate: were reminders to participate sent to potential participants to improve response rate?

4. Could people forward the survey to other potential respondents? This can sometimes bias the response rate.

5. Analyses: From the survey, I understand that the answers regarding the number of days are continuous numerical values (i.e. respondents indicated the number of days they would recommend), not a multiple-choice question (e.g. 7, 10, 14 days). Consequently, I’m not sure to understand why the main outcome is presented as “the proportion of respondents who recommend >=10 days of antibiotics” (page 8, line 187). Presenting the results in this manner would be expected if the question had been a multiple choice question, but considering the type of data available, I would expect these results to be presented as a median number of days with IQR. Saying that “65% recommend at least 10 days of treatment for pneumonia” is not the same (and not as informative) as saying that the median duration is 10 days (IQR, 7-10). Actually, the former sentence gives the impression that the treatment duration recommendation is longer than it really is (looking at the violin chart, 50% of respondents recommend 10 days or less of treatment). This comment would also apply to the abstract.

6. Analyses: the authors state that they conducted a multivariable linear regression analysis using specialty, country, years of practice and ASP practice as independent variables. However, when I look at table 3 and read the discussion section (page 18, lines 329-332), I have the impression that only bivariate (univariate) analyses were performed. Indeed, conducting multiple univariate analyses is not the same thing as conducting a multivariate analysis. Please clarify whether multiple variables were entered simultaneously into a single model to identify independent predictors after adjusting for covariates.

7. As an additional analysis, I would be curious to know whether there is cross-correlation in terms of treatment duration recommendation for a given respondent. For example: is a respondent who recommend a longer treatment for a given infection also more likely to recommend a longer treatment for another type of infection? Identifying cross-correlation would allow to identify subgroups of “long-prescribers” and “short-prescribers”.

8. Table 3. A footnote to explain how to interpret the B-coefficient would be helpful.

9. Considering the small sample size and the multiple comparisons that were conducted between groups, I wonder whether the p-value should be adjusted for multiple comparisons and be lower than 0.05. This could prevent the identification of spurious findings.

Reviewer #2: In this study, Dr. Pong and colleagues aimed to understand typical practice patterns surrounding duration of therapy for bacteremia in pediatric patients. They designed an online survey that was administered to pediatric intensivists, nurse practitioners, ID physicians, and pharmacists. They report interesting data that should motivate RCTs in this area. Some minor comments below.

Abstract

Lines 59-60: This is a little unclear. I think that it would be helpful avoid the nested parentheses and perhaps use brackets instead. Also, make it more clear that the first set of numbers refers to lack of source control with intra-abdominal infections and second set of numbers refers to patients with central line infections

Lines 60: It would be helpful to clarify why there is a range here. Perhaps “73-95%, depending on source of bacteremia”

Lines 63-64: I don’t think that this concept of implicit versus explicit equipoise is going to be familiar to most readers and so it either warrants a brief explanation or just remove it. It would be important to point out how the results presented in the abstract support this statement.

Introduction:

Why transition in terminology between bacteremia (Abstract) and then BSIs (the rest)?

Results:

Lines 188-190: Why the different denominators for each infectious syndrome, and why none equal to 136? Participants had the option of not answering questions?

Table 3: It would be helpful to note that the beta-coefficient and CI represent days of therapy

Discussion:

Lines 327-335: Could consider omitting this paragraph. Identifying differences in treatment duration between the various subgroups is challenging given the overall low numbers in the subgroups and the different compositions of the subgroups (e.g., more ID providers in the New Zealand/Australia group).

Reviewer #3: This manuscript nicely describes the variability in AB prescription patterns for SBI in children. As expected, there is a wide range of treatment duration as beautifully showed in the violin plots, but the general duration is long (10-14 days) and to me always intriguing, durations are usually 5, 7, 10, 14 or 21 days, never a number in between.

The manuscript is very clear and well written. One question that came into my mind is how are the prescription patterns related to the national protocols? Did authors look at national protocols for paediatric SBI and compared those protocols with the answers of the respondents? This probably would be interesting to add although there is also variability within a country.

The manuscript is a good base for starting of a trial into duration of SBI treatment. I am wondering if there is any place for IV-oral switch in those SBIs (see review DOI: 10.1016/S1473-3099(16)30024-X) , this is nowhere mentioned by the authors and might be addressed in the discussion section.

Specific comments:

The introduction is very well written and covers the most important aspects of questions concerning antibiotic use. The review of McMullan could be referred to (DOI: 10.1016/S1473-3099(16)30024-X)

Line 228/229: is this sentence correct? Is should probably read longer durations after no removal of catheter?

Table 3: very interesting an ASP present does not lead to differences in treatment duration - authors could highlight this in the discussion.

6. PLOS authors have the option to publish the peer review history of their article (what does this mean?). If published, this will include your full peer review and any attached files.

Reviewer #1: No

Reviewer #2: No

Reviewer #3: **Yes: **G.A. Tramper

---

## [Author Response · Author response to Decision Letter 0]

3 Jul 2022

Dear Dr. Vadivelu,

Thank you for your consideration of our manuscript, entitled “Antibiotic treatment duration for bloodstream infections in critically ill children—a survey of pediatric infectious diseases and critical care clinicians for clinical equipoise” for publication in PLOS ONE. We are pleased to hear that your Editorial team believes our manuscript is of interest to your readership. We greatly appreciated your feedback and reviewer comments and have strived to incorporate the suggestions into the manuscript. We believe the paper has been strengthened in the process. The detailed responses to the Editor and Reviewers’ comments are provided below.

Sincerely,

Sandy Pong

Corresponding Author:

Sandra Pong, PharmD

Clinical Pharmacist 

The Hospital for Sick Children

Department of Pharmacy

555 University Avenue

Toronto, Ontario M5G 1X8

Canada

sandra.pong@sickkids.ca

Journal Requirements:

In your Data Availability statement, you have not specified where the minimal data set underlying the results described in your manuscript can be found. 

We have added our dataset to “Supplementary materials” (S7 Dataset).

We note that you have included the phrase “data not shown” in your manuscript. Unfortunately, this does not meet our data sharing requirements. PLOS does not permit references to inaccessible data. We require that authors provide all relevant data within the paper, Supporting Information files, or in an acceptable, public repository. 

We have added these data to “Supplementary materials” (S2-S6 Tables).

Please review your reference list to ensure that it is complete and correct. Any changes to the reference list should be mentioned in the rebuttal letter that accompanies your revised manuscript. 

We have reordered and added additional references according to changes in the main body of the manuscript. No references were deleted or changed. 

Reviewer #1: Review of article entitled “Antibiotic treatment duration for bacteremia in critically ill children--a survey of pediatric infectious diseases and critical care clinicians for clinical equipoise” by S. Pong et al.

Major comments

Interesting study. The project is well designed and the results are presented clearly. The use of violin plots is judicious. I have the following comments to help improve the manuscript.

1. Abstract: could we indicate how many scenarios were presented? Perhaps mentioning that the survey focused on 5 common pediatric scenarios.

We have revised this sentence to say “…five common pediatric-based case scenarios of bacteremia.” (page 2, line 47-48)

2. Respondents: could an individual answer the survey twice?

To maintain anonymity of survey participants, we did not track individual respondents and could not identify if an individual answered the survey more than once, but we do not expect that a survey respondent would complete this survey multiple times. 

3. Invitation to participate: were reminders to participate sent to potential participants to improve response rate?

Yes, two reminder email invitations were sent to participants after the initial invite. We have added this clarification to the Methods section. (page 6, line 124-125)

4. Could people forward the survey to other potential respondents? This can sometimes bias the response rate.

We did not request participants to forward the survey to other potential respondents. There were instances when invitees inquired if they could forward the survey to colleagues. In those cases, we asked if they could let us know the number of additional colleagues who they were going to send the survey to, or if they could send us their email addresses so that we could contact them directly with an invitation to complete a survey. However, there was no mechanism in place that would prevent someone from forwarding the survey without our knowledge. 

5. Analyses: From the survey, I understand that the answers regarding the number of days are continuous numerical values (i.e. respondents indicated the number of days they would recommend), not a multiple-choice question (e.g. 7, 10, 14 days). Consequently, I’m not sure to understand why the main outcome is presented as “the proportion of respondents who recommend >=10 days of antibiotics” (page 8, line 187). Presenting the results in this manner would be expected if the question had been a multiple choice question, but considering the type of data available, I would expect these results to be presented as a median number of days with IQR. Saying that “65% recommend at least 10 days of treatment for pneumonia” is not the same (and not as informative) as saying that the median duration is 10 days (IQR, 7-10). Actually, the former sentence gives the impression that the treatment duration recommendation is longer than it really is (looking at the violin chart, 50% of respondents recommend 10 days or less of treatment). This comment would also apply to the abstract.

We used ≥10 days as a cut-off to arbitrarily define longer duration treatment, which has also been done in previous work by our group in adult populations. In our analyses of whether respondents would be willing to enrol patients into a trial of shorter vs. longer antibiotic treatment, we also used this dichotomy to determine the proportion who had also recommended ‘longer’ treatment durations in the case scenarios.

We feel this dichotomization also facilitates the cross-correlation analyses that were suggested by Reviewer #1 to assess if recommendations for longer treatment duration for a given infectious syndrome are associated with longer treatment duration for another infectious syndrome. 

We have added that we defined ‘longer duration treatment” as ≥10 days and referenced the previous use of this in Daneman et al. Crit Care Med 2016;44:256-264. (page 8, line 161-162)

6. Analyses: the authors state that they conducted a multivariable linear regression analysis using specialty, country, years of practice and ASP practice as independent variables. However, when I look at table 3 and read the discussion section (page 18, lines 329-332), I have the impression that only bivariate (univariate) analyses were performed. Indeed, conducting multiple univariate analyses is not the same thing as conducting a multivariate analysis. Please clarify whether multiple variables were entered simultaneously into a single model to identify independent predictors after adjusting for covariates.

Yes, the multiple variables (specialty, country, years since graduation and antimicrobial stewardship program) were entered simultaneously into single models for each infectious syndrome (pneumonia, skin/soft tissue, urinary tract, intra-abdominal and central vascular catheter). In the case of intra-abdominal infections, we included source control (partial/not drained vs. drained) as an additional variable in the model. In the case of central vascular catheter-associated infections, we included source control (catheter removed vs. not removed) and pathogen type as additional variables in the model. These models are summarized in Table 3. 

We have clarified this by specifying these are multivariable regression models in the heading of Table 3. “Table 3. Multivariable regression models for predictors of recommended treatment duration for different infectious syndromes.” (page 14, line 261-262)

7. As an additional analysis, I would be curious to know whether there is cross-correlation in terms of treatment duration recommendation for a given respondent. For example: is a respondent who recommend a longer treatment for a given infection also more likely to recommend a longer treatment for another type of infection? Identifying cross-correlation would allow to identify subgroups of “long-prescribers” and “short-prescribers”.

We have added “Supplement Table 6 (S8 Table)” showing pairwise comparisons of respondents’ recommendations for longer treatment duration for bacteremia due to pneumonia, skin/soft tissue infection, urinary tract infection, intra-abdominal infection (drained, partial/not drained) and central venous catheter-associated infection, due to E. coli, as an example organism (catheter removed, catheter not removed). 

8. Table 3. A footnote to explain how to interpret the B-coefficient would be helpful.

We have added a footnote to Table 3: “Adjusted �-coefficient for each predictor variable represents the change in number of days of antimicrobial therapy for the variable relative to the reference predictor variable.” (page 16, line 264-265)

9. Considering the small sample size and the multiple comparisons that were conducted between groups, I wonder whether the p-value should be adjusted for multiple comparisons and be lower than 0.05. This could prevent the identification of spurious findings.

We have made Bonferroni correction to the p-values in Tables 2 and 3. 

Table 2—in the comparison of treatment duration between specialities, the adjusted p-value threshold is 0.02. (page 12, line 244)

Table 3—in the multivariable regression models: for ‘Specialty’, the adjusted p-value threshold is 0.02, for ‘Years since graduation’ the adjusted p-value threshold is 0.005 and for ‘Pathogens’, the adjusted p-value threshold is 0.002. (page 16, line 266)

Reviewer #2: In this study, Dr. Pong and colleagues aimed to understand typical practice patterns surrounding duration of therapy for bacteremia in pediatric patients. They designed an online survey that was administered to pediatric intensivists, nurse practitioners, ID physicians, and pharmacists. They report interesting data that should motivate RCTs in this area. Some minor comments below.

Abstract

Lines 59-60: This is a little unclear. I think that it would be helpful avoid the nested parentheses and perhaps use brackets instead. Also, make it more clear that the first set of numbers refers to lack of source control with intra-abdominal infections and second set of numbers refers to patients with central line infections

We have revised the sentence to clarify: “In multivariable linear regression analyses, lack of source control was significantly associated with longer treatment durations (+5.2 days [95% CI: 4.4-6.1 days] for intra-abdominal infections and +4.1 days [95% CI: 3.8-4.4 days] for central vascular catheter-associated infections).” (page 2, line 60-63) 

Lines 60: It would be helpful to clarify why there is a range here. Perhaps “73-95%, depending on source of bacteremia”

We have revised the sentence: “Most clinicians (73-95%, depending on the source of bacteremia) would be willing to enrol patients into a trial of shorter versus longer antibiotic treatment duration.” (page 2, line 63-65)

Lines 63-64: I don’t think that this concept of implicit versus explicit equipoise is going to be familiar to most readers and so it either warrants a brief explanation or just remove it. It would be important to point out how the results presented in the abstract support this statement.

We have removed references to implicit and explicit equipoise in the abstract and replaced with: “There is practice heterogeneity in self-reported treatment duration recommendations among clinicians. Treatment durations were similar across different infectious syndromes. Under appropriate clinical conditions, most clinicians would be willing to enrol patients into a trial of shorter versus longer treatment for common syndromes associated with bloodstream infections.” (page 3, line 67-71)

Introduction:

Why transition in terminology between bacteremia (Abstract) and then BSIs (the rest)?

We have replaced ‘bacteremia’ with ‘bloodstream infections’ in the abstract and the manuscript title. 

Results:

Lines 188-190: Why the different denominators for each infectious syndrome, and why none equal to 136? Participants had the option of not answering questions?

The different denominators were the result of a few participants not answering every question. 

Table 3: It would be helpful to note that the beta-coefficient and CI represent days of therapy

We have added clarification as a footnote to Table 3: “Adjusted �-coefficient for each predictor variable represents the change in number of days of antimicrobial therapy for the variable relative to the reference predictor variable.” (page 16, line 264-265)

Discussion:

Lines 327-335: Could consider omitting this paragraph. Identifying differences in treatment duration between the various subgroups is challenging given the overall low numbers in the subgroups and the different compositions of the subgroups (e.g., more ID providers in the New Zealand/Australia group).

Although we had small overall numbers, we feel that it is worthwhile to mention the subgroups with respect to the potential problems of confounding given that respondents in Australia and New Zealand were primarily ID clinicians. 

We have added a statement to acknowledge that the overall numbers in the subgroups are small: “However, the overall number of respondents in our subgroups were small and this difference among countries may be confounded by survey respondents in Australia and New Zealand being primarily ID clinicians.” (page 19, line 345-347)

Reviewer #3: This manuscript nicely describes the variability in AB prescription patterns for SBI in children. As expected, there is a wide range of treatment duration as beautifully showed in the violin plots, but the general duration is long (10-14 days) and to me always intriguing, durations are usually 5, 7, 10, 14 or 21 days, never a number in between.

The manuscript is very clear and well written. One question that came into my mind is how are the prescription patterns related to the national protocols? Did authors look at national protocols for paediatric SBI and compared those protocols with the answers of the respondents? This probably would be interesting to add although there is also variability within a country.

We did not compare survey responses with national protocols. We expected significant practice variability to exist across institutions and across countries, so comparisons of national protocols may not be meaningful. There may also be greater reliance on provincial/state or local regional protocols rather than national level protocols.

The manuscript is a good base for starting of a trial into duration of SBI treatment. I am wondering if there is any place for IV-oral switch in those SBIs (see review DOI: 10.1016/S1473-3099(16)30024-X) , this is nowhere mentioned by the authors and might be addressed in the discussion section.

We have added to our discussion: “In addition, earlier switch from intravenous to suitable oral antibiotics with patient defervescence and clinical improvement is another potential strategy to optimize treatment duration and reduce antimicrobial overuse,” and referenced this systematic review and guideline. (page 18, line 323-326)

Specific comments:

The introduction is very well written and covers the most important aspects of questions concerning antibiotic use. The review of McMullan could be referred to (DOI: 10.1016/S1473-3099(16)30024-X)

We have included a reference to this systematic review and guideline. (page 4, line 79) 

Line 228/229: is this sentence correct? Is should probably read longer durations after no removal of catheter?

This sentence is correct, but we agree that it is unclear as written. The meaning of “shorter” is in comparison to durations recommended by critical care clinicians and pharmacists. We have revised the sentences in this section to make the meaning clearer.

“For central vascular catheter-associated bacteremia with source control (infected catheter removed), ID physicians recommended significantly shorter treatment durations than critical care clinicians and pharmacists for E. faecalis and coagulase negative staphylococci infections. When there was no removal of infected catheters, ID physicians recommended significantly shorter durations than critical care clinicians and pharmacists for all pathogens, except for S. aureus, K. pneumoniae and P. aeruginosa infections (Table 2).” (page 12, line 233-238)

Table 3: very interesting an ASP present does not lead to differences in treatment duration - authors could highlight this in the discussion.

We have added to our discussion: “The presence of an institutional antimicrobial stewardship program was not associated with a significant difference in treatment duration recommendations made by survey respondents.” (page 19-20, line 350-352)

---

## [Editor Report · Decision Letter 1]

12 Jul 2022

Antibiotic treatment duration for bloodstream infections in critically ill children—a survey of pediatric infectious diseases and critical care clinicians for clinical equipoise

PONE-D-22-04577R1

Dear Dr Pong,

We’re pleased to inform you that your manuscript has been judged scientifically suitable for publication and will be formally accepted for publication once it meets all outstanding technical requirements.

Kind regards,

Jamunarani Vadivelu

Academic Editor

PLOS ONE
---

## [Editor Report · Acceptance letter]

18 Jul 2022

PONE-D-22-04577R1 

Antibiotic treatment duration for bloodstream infections in critically ill children—a survey of pediatric infectious diseases and critical care clinicians for clinical equipoise 

Dear Dr. Pong:

I'm pleased to inform you that your manuscript has been deemed suitable for publication in PLOS ONE. Congratulations! Your manuscript is now with our production department. 

Kind regards, 

on behalf of

Dr. Jamunarani Vadivelu 

Academic Editor

PLOS ONE